

# What's the effectiveness of stocking actions in small creeks? The role of water discharge behind hatchery trout downstream movement

Stefano Brignone[1], Vanessa De Santis[1], Tiziano Putelli[2],
Christophe Molina[2], Armando Piccinini[3], Richard A. Carmichael[3] and
Pietro Volta[1]

[1] National Research Council CNR-IRSA, Verbania, Italy
[2] Ufficio della caccia e della pesca, Repubblica e Cantone Ticino, Bellinzona, Canton Ticino, Switzerland
[3] Biomark, Boise, USA

Corresponding author
Stefano Brignone, stefano.brignone@irsa.cnr.it

## ABSTRACT

Fish stocking to enhance freshwater fisheries or to improve the conservation status of endangered fish species is a common practice in many countries. Little is known, however, of the effectiveness of these practices in spite of the high efforts and investments required. The movement of subadult/adult hatchery-released brown trout *Salmo trutta* L. was studied by passive telemetry in a small tributary of Lake Lugano (*i.e.*, Laveggio Creek, Canton Ticino, Switzerland). Hatchery fish, together with some resident wild individuals sampled during electrofishing surveys, were tagged with Passive Integrated Transponders (PIT) tags. Hatchery fish were released upstream and downstream a submersible monitoring antenna, which was anchored to the streambed in a pass-over orientation. The number of hatchery fish detected daily by the antenna (divided between fish released upstream and downstream the antenna) was analyzed in relation to the daily water discharge, to search for similar patterns in their fluctuation over time. Only the movement of fish released upstream the antenna displayed a significant relationship with water discharge, with the highest number of fish detected during periods of high-water flow, occurring after heavy rains. High-water discharge events had a significant role in hatchery trout downstream movement in our study site, likely acting as a driver for the downstream migration to Lake Lugano. Such events contributed to the poor effectiveness of stocking actions in this small tributary, providing further evidence against stocking strategies based on subadult/adult fish.

## INTRODUCTION

Brown trout *Salmo trutta* L. is one of the fish species with the highest economic value, both for professional and recreational fisheries (*ICES, 2016*; *Schwinn et al., 2017*).
Hatchery-reared brown trout have been used for a long time to enhance and/or maintain artificial populations in fresh waters (*Einum & Fleming, 2001*; *Aprahamian et al., 2003*; *Flowers et al., 2019*). However, hatchery fish face several problems compared with wild conspecifics, such as reduced fitness and feeding efficiency, and high mortality (*Waples, 1999*; *Brannon et al., 2004*; *Araki, Cooper & Blouin, 2007*). This is usually due to rearing conditions presenting high densities, unnatural food sources and absence of external stimuli, like predation and variations in water flow (*Fernö & Järvi, 1998*; *Aarestrup et al., 2005*; *Pedersen, Koed & Malte, 2008*).

Millions of hatchery-reared trout are still released into creeks/streams every year (*Waples, 1999*; *Aprahamian et al., 2003*; *Brockmark & Johnsson, 2010*; *Polgar et al., 2022*) despite their fate and survival being largely unknown after the release into the wild. Furthermore, a few studies have investigated the environmental variables behind the movements of hatchery-reared brown trout following their release into water courses, especially in creeks and little streams (*Helfrich & Kendall, 1982*; *Pedersen, Dieperink & Geertz-Hansen, 2003*; *Aarestrup et al., 2005*).

Fish movement is influenced by a variety of factors. One of the main driving forces for large-scale fish migrations is the search for suitable spawning habitats (*Northcote, 1992*; *Larinier, 2000*; *Young et al., 2010*; *Alò et al., 2021*). However, fish can move along streams and rivers for several other reasons, such as to seek better feeding grounds or better water conditions, including more suitable thermal conditions (*Hughes, 1998*; *Gowan & Fausch, 2002*; *Hughes et al., 2003*).

Movement is not always a volunteer action, as fish can be displaced downstream because of high-water discharge, and this is especially true for juveniles, less capable to withstand high-water flows compared to adults (*Jowett & Richardson, 1989*; *Hayes, 1995*; *Nislow et al., 2002*).

Contrastingly, periods of high-water flow may also favor upstream movements allowing fish to take advantage of temporary passages created during high-flow conditions and, in general, to overcome in-river barriers more easily (*Jonsson, 1991*; *Jowett & Richardson, 1994*; *Dedual & Jowett, 1999*; *Schwartz & Herricks, 2005*).

In many salmonid species water discharge variation, combined with other environmental factors such as water temperature and photoperiod, regulates the downstream migration from nursery grounds (*Elliott, 1994*; *Klemetsen et al., 2003*; *Byrne et al., 2004*). Fish move to better feeding areas or to areas with better environmental conditions, such as lakes or seas/oceans (*Klemetsen et al., 2003*), to maximize fitness (*Olsson & Greenberg, 2004*).

This migration is part of the smoltification process, which is preceded by a series of morphological, physiological and behavioral changes in juveniles, and starts as they reach a minimum size, depending on the salmonid species (*Thorstad et al., 2011*).

Brown trout smolts, for instance, initiate their downstream migration a few years after hatching, usually as 2- and 3-year-old individuals. However, hatching can also take place in 1- to 8-year-old fish/individuals (*Elliott, 1994*; *Jonsson et al., 1999*; *Klemetsen et al., 2003*).

Despite the migratory potential of the species, stocking subadult and adult fish is still considered as a useful method to increase biomass of catchable fish both in the short and
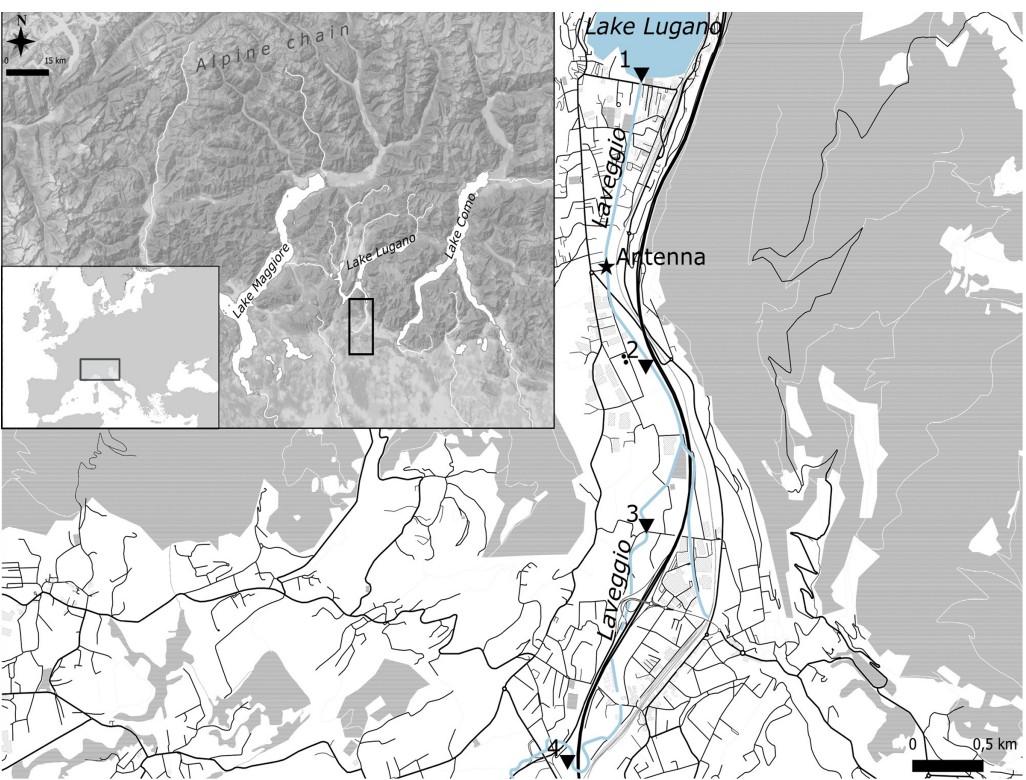

**Figure 1 Study area.** Black triangles are the four stations where marked hatchery trout were released.

long term. To test the effectiveness (*i.e.*, the short-term availability of catchable fish) of brown trout stocking with subadult and adult fish in a small creek, we analyzed the movement of hatchery-reared brown trout released in Laveggio Creek (Lake Lugano catchment, Canton Ticino, Southern Switzerland), assessing the role exerted by water discharge in altering their post-release residence time and abundance.

We hypothesize that water discharge plays an important role in fish movement, either as a physical stimulus behind hatchery trout migration towards lake habitats (active movement), or as a physical factor forcing hatchery fish to move downstream, due to their scarce attitude to sustained swimming during high water discharge events (passive movement).

## MATERIALS AND METHODS

### Study area

Laveggio is a small creek located in the southern side of the Alps (45°54′N 8°58′E), at the Swiss–Italian border within the Lake Lugano basin. Lake Lugano is a eutrophic lake with a surface area of 49 km$^2$ (Fig. 1). Laveggio Creek has a spring origin, flowing for 11 km; the streambed width ranges from 4 to 8 m, with many human-altered and channelized sections, except for some of its uppermost portions (https://map.geo.admin.ch). Located in an area with high rainfall (mean daily precipitation: 4.0 ± 10.7 mm, max daily

precipitation: 82.5 ± 11.2 mm, last 5 years data), this creek can undergo dramatic changes in its discharge rate, reaching more than 10 m$^3$ s$^{-1}$ (mean daily discharge: 1.1 ± 1.2 m$^3$ s$^{-1}$).

Water temperature varies significantly during the year, peaking in the summer (~27 °C) and reaching its coldest point in the winter (~6 °C).

Precipitation and water temperature data were taken from the Environmental Observatory of Switzerland (https://www.oasi.ti.ch/web/dati/idrologia.html).

The fish community is made of brown trout *S. trutta*, European river lamprey *Lampetra fluviatilis* (L.), European perch *Perca fluviatilis* L., roach *Rutilus rutilus* (L.), Italian chub *Squalius squalus* (Bonaparte, 1837) and the Italian dace *Telestes muticellus* (Bonaparte, 1837).

On the streambed of Laveggio, around 1.2 km upstream from the confluence with Lake Lugano (Fig. 1), we positioned two high-density polyethylene (HDPE) antennas (3 m each; Biomark, Boise, Idaho, USA), which were lined up to cover the entire width of the streambed, forming a singular flat-bed PIT monitoring system. The antennas were anchored to the streambed of Laveggio, at the end of a large pool in approx. 30 cm of water, to make sure to detect fish during high-water events.

This system usually registers the passage of PIT tagged fish moving up to 10 m s$^{-1}$ and within a range up to 0.9 m, although the latter can vary according to the physico-chemical properties of waters and to the background noise (*e.g.*, electromagnetic interference from a nearby-electrified railway). Once fish enter the electromagnetic field generated by the antennas, the implanted PIT tag is charged and can transmit its unique identification code.

The PIT tag antenna system together with its recording station (Biomark Master Controller, Boise, Idaho, USA), was installed and anchored to the streambed at the end of August 2020.

## Fish sampling

Fish used in this study came from the fish hatchery managed by the Office of hunting and fishing of Canton Ticino, Bellinzona (Switzerland). After eggs hatched, trout were reared under natural light conditions and fed with commercial pellet.

Stocking with subadult and adult fish in Laveggio Creek is forbidden by law and, thus, the stocking is usually carried out using eggs and fry. However, to study the movement of trout in Laveggio, we were allowed to mark and release 998 subadult/adult hatchery-reared trout during autumn 2020, and 520 during autumn 2021.

PIT tags (Biomark APT12, Advanced Performance Tag, 12 × 2 mm, ISO Standard, 134.2 kHz) were implanted in the intraperitoneal cavity through a Biomark MK25 implanter and pre-loaded single-use needles to minimize fish stress and possible disease transmission and to reduce the healing time. Fish underwent a light eugenol anesthesia during handling and tagging operations, while feeding was stopped 2 days prior to the marking day. All trout remained in the hatchery, in a dedicated tank for at least 10 days after tagging and before release to monitor the healing of the implant incision and to allow a full recovery. Feeding resumed 24 h after the tagging day. No mortality was observed during the 10 days of recovery time.

Fish released in 2020 were equally divided between fish of age $1^+$ (total length range: 15–20 cm) and $2^+$ (total length range: 20–30 cm), whereas all fish released in 2021 were of age $1^+$. Hatchery trout were marked and released at the end of October 2020 and 2021 at four different sectors of the creek, one sector close to the confluence with Lake Lugano, downstream of the antenna, and three sectors upstream of the antenna (Fig. 1).

After a few days of data monitoring, some of the fish seemed to pause in the proximity of the antenna; this location was probably used as temporary feeding area and refuge. A delay of 15 s between each recording was necessary to avoid memory overload.

During the study period, in order to search for marked trout, we conducted six electrofishing surveys using a Hans-Grassl EL64GII electrofisher, with settings of 7,000 W, 300 V. Because of the study area extension, we had to focus on different portions of the creek in each electrofishing survey: between sector 3 and 4 during the first survey (January 2021), near sector 1 and 3 during the second (February 2021), near sector 2 and 3 during the third (first half of April 2021), near the antenna and between the sector 3 and 4 during the fourth (second half of April 2021), between sector 3 and 4 during the fifth (June 2021), and near the antenna during the sixth survey (December 2021). During three surveys (first and second half of April and December 2021), 226 wild fish were captured, measured (total length and total weight) and marked with PIT tags before being released back to their site of capture.

Wild trout movement data was used to look for possible differences between hatchery and wild trout response to high water discharge events, although the number of wild trout marked was not comparable with hatchery conspecifics.

The experiment was conducted with permission of the 'Dipartimento della sanità e della socialità-Esperimenti su animali', authorization number TI-27-2020 (Supplemental Materials).

### Data collection

Fish passages started to be recorded by the antenna from October 29th, 2020, and terminated on April 30th, 2022, the last date in which hatchery trout were detected by the antenna. The main variable measured was the Number of Fish Detected Daily (hereinafter referred as NFDD) by the antenna, which is the total amount of individuals registered by the antenna during one day. Hatchery fish were divided into two groups based on whether they were released upstream (NFDD upstream) or downstream (NFDD downstream) the antenna, in order to discriminate the role of high-water events.

To assess the relation between NFDD (upstream and downstream) and daily water discharge, hydrological data were retrieved from the Environmental Observatory of Switzerland (https://www.oasi.ti.ch/web/dati/idrologia.html). Hydrological data were recorded by a monitoring station positioned close to the PIT tag antenna (Riva San Vitale).

### Statistical analysis

NFDD released upstream and downstream and water discharge fluctuated over time, so they were analyzed implementing a time series analysis approach.

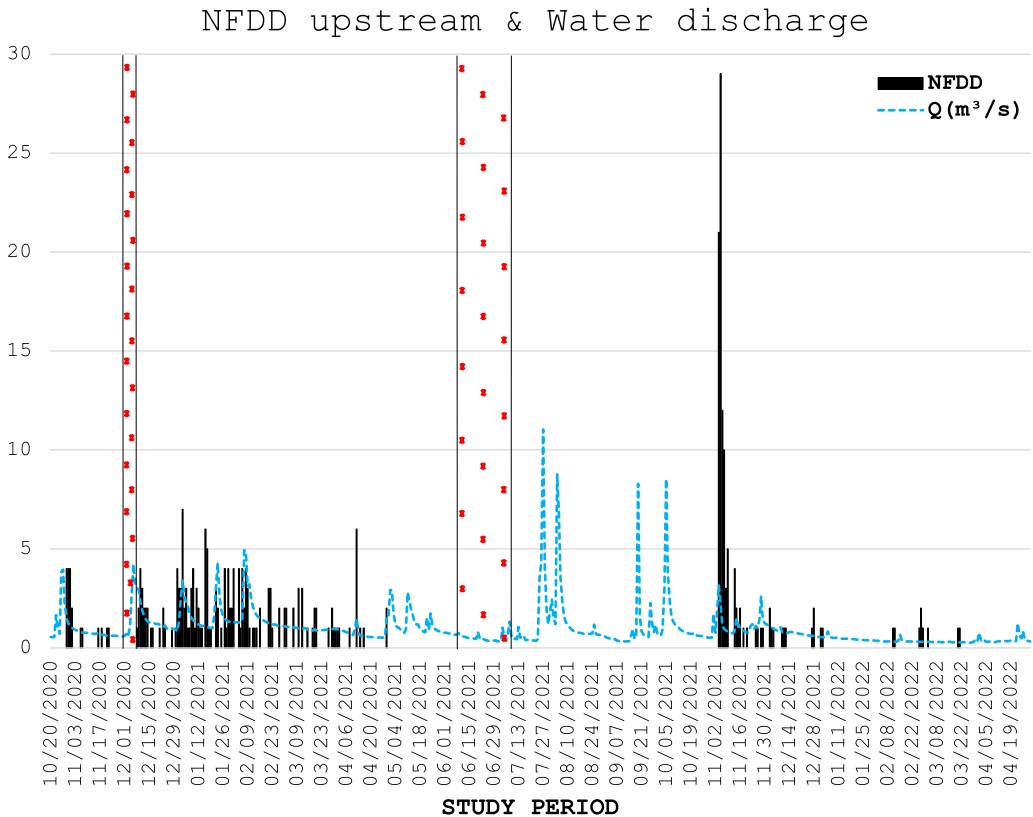

**Figure 2** **NFDD upstream & water discharge.** Trend comparison between number of fish detected daily (NFDD) released upstream the antenna and daily water discharge measured in Laveggio creek between October 2020 and April 2022. Dashed areas are the periods in which the antenna did not work.

The Autoregressive Integrated Moving Average (ARIMA) model was applied to analyze the structure of each variable. The optimal model was determined by automatic model selection based on the Akaike information criterion (AIC) function, included in the package "*forecast*" (*Hyndman et al., 2020*; *R Core Team, 2021*). The residuals of each resulting model were examined to check possible structures within each time series. In case a trend in the residual distribution of the time series is present, an adjustment for the estimated coefficients and standard errors is required to fit the generalized estimating equation (GEE) of generalized linear model, performed using the "*geepack*" package included in R software (*Højsgaard et al., 2016*).

Cross-correlation (CC) analysis was used to measure the similarity between time series and, in particular, to look for shifted patterns between water discharge time series and NFDD time series; this was done using the R package "*tseries*" (*Trapletti et al., 2015*).

## RESULTS

In total, 1,518 hatchery trout were PIT-tagged and released in Laveggio Creek. Among these trout, 222 fish were registered by the antenna at least once, which corresponded to only the 14.6% of the total amount of trout released in the creek. The highest number of fish registered by the antenna in a day was 38, one day after the release date in 2021.

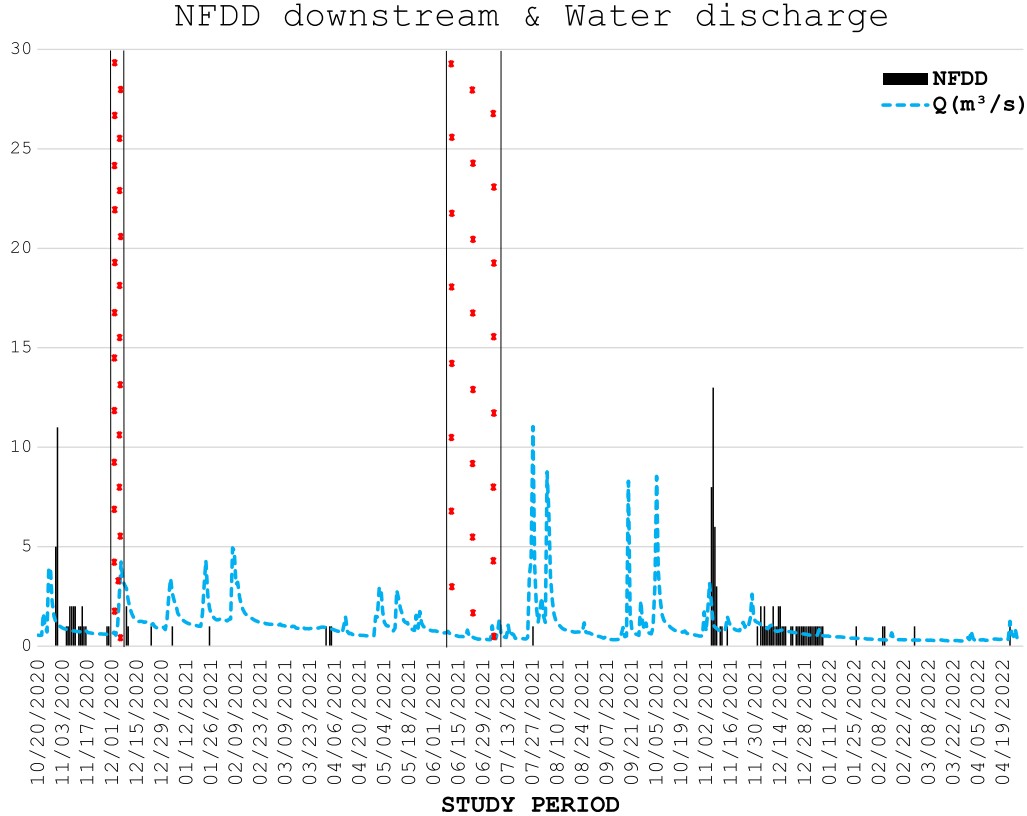

**Figure 3 NFDD downstream & water discharge.** Trend comparison between number of fish detected daily (NFDD) released downstream the antenna and daily water discharge measured in Laveggio creek between October 2020 and April 2022. Dashed areas are the periods in which the antenna did not work.

From November 29th to December 7th, 2020, the antenna stopped recording data because of memory overload, whereas from June 7th to July 9th, 2021, it stopped working for an electrical malfunction (Figs. 2, 3).

During the five electrofishing surveys conducted in the first half of 2021, only nine marked trout were recaptured: six in January, one in February, two in the first half of April and zero during the last two surveys in the second half of April and in June. During the survey of June 2021 23 marked wild trout were recaptured. During the electrofishing survey of December 2021, 1 month after the second release of hatchery trout in Laveggio Creek, only four hatchery trout were recaptured, together with one marked wild trout.

Electrofishing efficiency was supposed to be high, since together with the few hatchery trout found during these surveys, we were able to sample several wild trout along Laveggio Creek, most of them under-size for PIT tag marking and therefore not considered further.

During three electrofishing surveys, 226 wild trout were marked with PIT tags (75 and 139 fish in the first and second part of April 2021, respectively, and 12 in December 2021), with an average length of 18.7 ± 4.7 cm. The number of wild fish detected daily by the antenna is shown in Fig. 4.

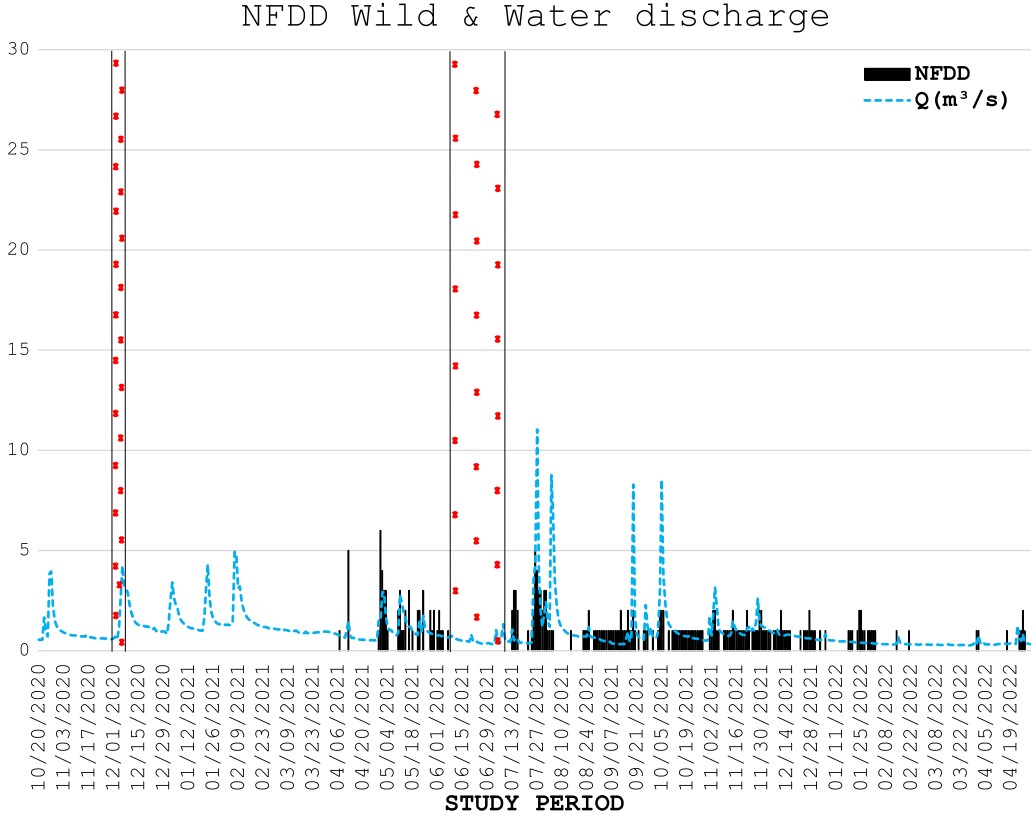

**Figure 4 NFDD wild & water discharge.** Trend comparison between number of wild fish detected daily (NFDD) and daily water discharge measured in Laveggio creek between April 2021 and April 2022. Dashed areas are the periods in which the antenna did not work.

Six of the wild fish detected by the antenna stationed in the proximity of the antenna for a very long time (two individuals for almost one year), suggesting that these trout particularly appreciated the antenna area which was potentially used as refuge/feeding ground.

Water discharge was on average $1.0 \pm 1.0$ m$^3$ s$^{-1}$ and exceeded $3.5$ m$^3$ s$^{-1}$ for 13 non-consecutive days, usually after short periods of heavy rains, and water levels remained high for several days after these events (Figs. 2, 3).

For each analyzed time series, ARIMA model outputs revealed that residuals were randomly distributed (Supplemental Materials). ACF and PACF did not show any relevant peak for the time series of NFDD released downstream (Fig. S2), whereas the ACF and PACF of NFDD released upstream and water discharge time series showed one or two peaks far from the zero, respectively (Figs. S1 and S3 for NFDD released upstream and water discharge series, respectively).

Since residuals of the ARIMA model were randomly distributed, with no evident structure, there was no need to adjust the estimated coefficient and standard error, and the generalized estimating equation (GEE) could be fitted.

**Table 1 Results of the generalized linear model analysis using the generalized estimating equation (GEE), performed between NFDD released upstream and water discharge; an asterisk (\*) indicates *P*-value < 0.05.**

Analysis of GEE parameter estimates

|  | Estimate | Std. error | *P*-value |
|---|---|---|---|
| (Intercept) | 0.227 | 0.105 | 0.031* |
| Water discharge | 0.33 | 0.141 | 0.02* |

**Table 2 Results of the generalized linear model analysis using the generalized estimating equation (GEE), performed between NFDD released downstream and water discharge; an asterisk (\*) indicates *P*-value < 0.05.**

Analysis of GEE parameter estimates

|  | Estimate | Std. error | *P*-value |
|---|---|---|---|
| (Intercept) | 0.1961 | 0.0389 | 0.00000046* |
| Water discharge | 0.0245 | 0.0232 | 0.29 |

In the generalized linear regression model obtained through the GEE, only NFDD released upstream displayed a significant relationship with water discharge (Table 1). No significant relationship was found between NFDD released downstream and water discharge (Table 2).

Between NFDD released upstream and water discharge, cross-correlation (CC) found a correlation at lag 1, suggesting a 1-day shift between the two time series (Fig. 5).

CC analysis highlighted a shifted pattern between NFDD released downstream and water discharge at lag 2, suggesting a 2-days shift between the two time series (Fig. 6). However, since there was no clear relation between these time series, this result could be misleading.

## DISCUSSION

Trout stocking into fresh waters has been a common practice for decades (*Brockmark & Johnsson, 2010*), but its effectiveness and utility to support local populations and/or local fishing is debated (*Halverson, 2008*; *Araki & Schmid, 2010*; *Hunt & Jones, 2018*; *Pinter et al., 2018*).

The life stage of fish used for stocking may vary depending on the final aims of the stocking activities (conservation and/or supplementation for recreational fishing), as well as on the hatchery dimensions and production capacity (*Cowx, 1994*; *Letcher & Terrick, 2001*). Stocking with eggs (Vibert boxes, cocooning) and with yolk-sac fry is one of the most used procedures for stocking (*Kirkland, 2012*). However, the use of juveniles and subadult fish is still implemented in many freshwater fisheries and considered a suitable system to artificially increase the productivity of water courses (*Baer, Blasel & Diekmann, 2007*). Nevertheless, the adverse effects of domestication on fish physiology and behavior

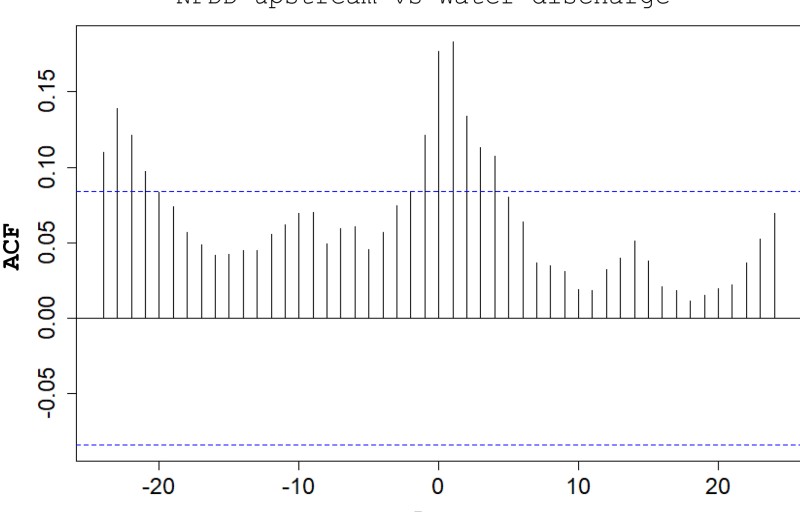

**Figure 5 NFDD upstream *vs* water discharge.** Cross-correlation (CC) analysis between NFDD released upstream and water discharge.               

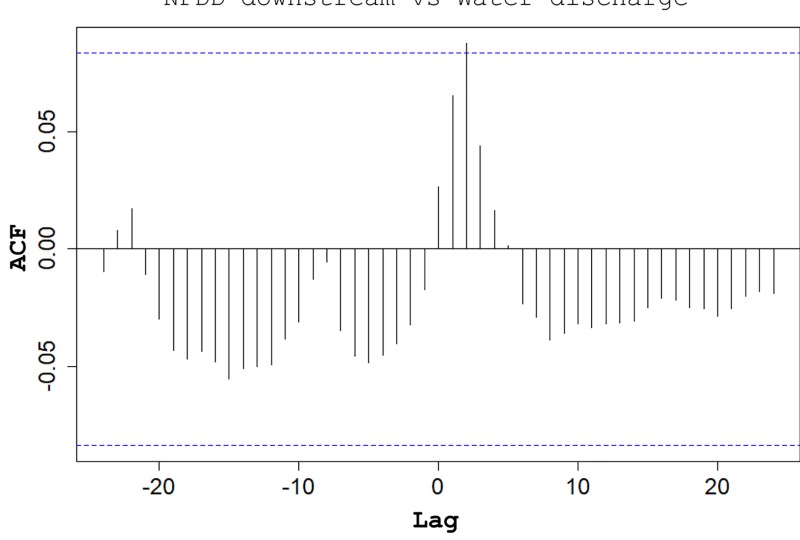

**Figure 6 NFDD downstream *vs* water discharge.** Cross-correlation (CC) analysis between NFDD released downstream and water discharge.          

are not often considered and fish mortality is usually underestimated (*Araki & Schmid, 2010*).

In this study we used PIT tag telemetry to monitor the movement of post-release subadult/adult hatchery trout in a small creek, a typical environment subject to stoking practices for fishery enhancement. The number of fish detected by the PIT tag antenna steadily decreased monthly and, only 6 months after both release dates, no more hatchery trout were detected by the antenna. The steep reduction in hatchery trout abundance after the release in wild ecosystems has been also observed in several other studies (*Näslund,*

*1992*; *Wiley et al., 1993*; *Pedersen, Dieperink & Geertz-Hansen, 2003*; *Pinter et al., 2018*), including also water courses of Canton Ticino (*Molina, 2019*) and Switzerland (*Gmünder & Friedl, 2002*).

Water discharge significantly influenced the movement of hatchery trout across the antenna in this study. The effects of water discharge strongly depended on the release site: peaks in water discharge were mirrored by peaks in fish passages only for fish released upstream the antenna, with 1-day lag after high-water events.

The relationship between fish movement and water discharge has been observed by many authors, in several fish species and environments. For example, *Tytler, Thorpe & Shearer (1978)* and *Thorpe et al. (1981)* observed a strong passive displacement in *Salmo salar* L. smolts, which were mainly transported to the ocean after high spring runoffs, which is one of the factors controlling their anadromous behavior (*Jonsson & Jonsson, 2009*). The downstream movement of smolts can also be a survival strategy, as highlighted by *Hansen & Jonsson (1985)*, where moving into the main current of the river allows smolts to avoid water sloughs and remnant pools. In contrast, *Brown, Power & Beltaoa (2001)* observed an opposite behavior: many fish take advantage of shelters in the creek margins, to minimize energy expenditure, avoid strong currents, and mitigate the risk of uncontrolled downstream dispersal. It has been observed that brown trout, in particular, move to side channels searching for refuges from peak flows (*Bunt et al., 1999*). These strategies enable fish to avoid physical injuries and getting trapped in shallow waters, where they would inevitably die as water dries up or because of anoxia (*Ortlepp & Mürle, 2003*). An alternative tactic used by some fish, described by *Brown, Power & Beltaoa (2001)*, is to take refuge in very large and deep pools during floods: these pools usually have a very low water velocity close to the bottom, which provides a suitable place where fish can endure flood events.

Water discharge variation is a well-known stimulus initiating brown trout smolt descend to lentic habitats, such as alpine lakes (*Klemetsen et al., 2003*); such habitats provide higher food availability and more favorable environmental conditions compared to nursery grounds (*Økland et al., 1993*; *Olsson et al., 2006*). The observed peaks in water discharge followed by peaks in fish moving downstream after a one-day lag suggest that these steep variations in water flow triggered the downstream movement of post-release hatchery trout.

Hatchery trout released in Laveggio Creek were brown trout of age 1+ and 2+, in which the smoltification process, although not assessed in this study, might have played a fundamental role in their downstream movement towards Lake Lugano, even after a few days/weeks from the release (*Flowers et al., 2019*).

Space and food competition with wild trout could also play an important role in the downstream migration of hatchery trout, since wild fish could induce hatchery trout to search for areas with lower intraspecific competition (low population density, higher food availability), *e.g.*, lake habitats (*Olsson et al., 2006*).

Another hypothesis behind the hatchery trout downstream movement, may be attributed to a passive transportation due to the high-water events occurred in Laveggio Creek. Laveggio is for a good portion of its lower course a channelized creek, with most of

its banks being completely human altered; furthermore, it exhibits few slow and deep pools where fish can take refuge. Therefore, trout may struggle to find any suitable cover on creek margins or in deep pools during high-water events. As stated before, spatial competition with established wild trout could also limit the number of available covers for post-release hatchery trout, through a prior-resident effect (*Deverill, Adams & Bean, 1999*; *Sánchez-González & Nicieza, 2021*).

Hatchery trout are more vulnerable to high-current conditions compared to wild trout (*Pedersen, Koed & Malte, 2008*): hatchery trout usually have reduced fin quality due to abrasion with tank sides and over-density conditions (*Drucker & Lauder, 2003*; *Ojanguren & Brana, 2003*). Lack of genetic diversity is also common in hatchery-reared fish, which, coupled with artificial selection, can favor the maintenance of stocks made by sedentary individuals that invest energy intake in growth rather than in movement (*Duthie, 1987*; *Molina, 2019*). Hatchery fish are usually bred under low flow rates, which may also reduce their activity levels, reducing activity of fatty acid oxidation enzymes in slow and fast muscle (*Johnston & Moon, 1980*).

Nevertheless, a good portion of the hatchery fish released upstream the antenna where registered again after the first passage across the antenna following a high-water event, so this passive transportation, if any, was limited to a little percentage of the released fish. Therefore, the results of the present study point mainly to the first hypothesis, for which water discharge acted mainly as a driver for the active downstream migration of the released hatchery trout.

These assumptions on the fate of hatchery trout along Laveggio Creek should take into account the antenna detection range and efficiency, which is a potential source of bias, quite difficult to estimate. In general, the detection efficiency of PIT tag antennas is not constant over time, and can rarely reach 100% (*Horton, Dubreuil & Letcher, 2007*; *Connolly et al., 2008*). Several factors can alter PIT tag antenna detection efficiency, such as tag orientation, water properties (*e.g.*, water salinity), tag collision and external electromagnetic interferences such as high-voltage power line (*Zydlewski et al., 2001*; *Aymes & Rives, 2009*; *Zentner et al., 2021*). However, Laveggio is a shallow creek, and the noise sources are limited (according to the noise levels measured by the recording station Biomark Master Controller), thus the detection efficiency during normal flows should be potentially very high. Nevertheless, high-water discharge events (such as the ones occurred during the study period) may have decreased the detection range and efficiency of the PIT tag antenna (*Zydlewski et al., 2006*; *Horton, Dubreuil & Letcher, 2007*). Hence, we could not exclude that there may have been other fish that moved over the antenna position, but which were not detected because of the reduced antenna detection range during periods of high-water flow. Yet, the low number of hatchery trout remained in the study area after a few months from the release was also pointed out by the results of five electrofishing surveys performed during the same period, in which only a few marked trout (13 trout, 0.9% of the total amount released into the creek) were detected. This low number could also be related to the low percentage of creek sampled during the electrofishing surveys, compared to the extension of the study area.

Despite our data suggest the importance of high-water discharge events behind the downstream movement of hatchery-reared brown trout (only fish released upstream the antenna were significantly related to water discharge), we cannot exclude the contribution of other factors in the apparent steep decrease of hatchery trout abundance. For instance, harvest by anglers can be very impactful on trout populations, especially in little creeks like Laveggio (*Almodóvar & Nicola, 2004*; *Wallace, 2010*; *Flowers et al., 2019*). Since anglers do not have the ability to detect or report interactions with PIT tagged fish, we were limited in our ability to assess angler exploitation and impact on movement and survival. However, the minimum size limit for trout harvest is set at 24 cm and at least 70% of the hatchery-reared fish were under this limit, so the role of fishing harvest on our results, if any, was limited.

Another important factor behind the drop in hatchery trout abundance from Laveggio could be linked to avian predation. This predation can be very effective and cause huge drops in trout abundance, and this is particularly relevant for hatchery fish (*Matkowski, 1989*; *Dieperink, Pedersen & Pedersen, 2001*). As stated before, hatchery fish are not used to sustained swimming and/or to predator avoidance, which, coupled with their more surface-oriented behaviors, makes them more vulnerable to bird predation than wild conspecifics (*Maynard, Flagg & Mahnken, 1995*; *Dieperink, Pedersen & Pedersen, 2001*). The collection of a PIT tag inside the stomach of a dead grey heron *Ardea cinerea* (L.) along the Laveggio Creek banks (C. Molina personal communication, 2021) could suggest a hidden effect of waterbirds on hatchery trout.

These factors, coupled with the downstream movement following high-water events, may help to explain the low percentage of recaptured trout during the electrofishing surveys, as well as the low percentage of fish detected by the antenna (0.9% and 14.6% of the marked hatchery trout, respectively), suggesting that most of the released trout died or moved downstream undetected by the antenna. Unfortunately, it was not possible to discriminate between these two hypotheses, and this aspect should be considered in future studies.

Wild trout movements seemed to be less influenced by high water events than hatchery trout. Some of the wild fish marked were detected by the antenna for the entire duration of the study period, despite the repeated high-water events particularly occurring in summer and early autumn 2021. However, comparisons between wild and hatchery trout should be taken with caution since data on wild fish were scarce, both for the low number of wild fish marked and for the low number of wild fish detected by the antenna; besides, wild trout were marked in different time period than hatchery trout. For these reasons any further assumptions on the resilience of wild trout and their differences with hatchery trout behavior in Laveggio Creek are limited.

In conclusion, our results suggest that stocking for fishery enhancement in creeks like Laveggio should probably be adjusted and moved toward different methods, since the traditional stocking activities with subadult/adult trout is not effective. We highlighted how fragile the stability of artificial populations made of hatchery trout is in this type of environment, and how easily different biotic and abiotic factors can disrupt this stability.

We cannot exclude the usefulness of these stocking actions on the long-term period, since hatchery trout moved downstream may potentially return back in Laveggio Creek as spawners, enhancing the natural stock of this freshwater system. The PIT tag antenna monitoring system used in this study will be maintained for the next years and it will contribute to shed new light on this possible hidden aspect.

## CONCLUSIONS

This study showed how the effectiveness of stocking practices using subadult and adult hatchery brown trout is limited by downstream movements toward lentic connected systems, eased by high-water discharge events. Peaks in water discharge mirrored peaks in PIT-tagged hatchery trout detections registered by a PIT tag antenna system. The number of fish detected by the antenna decreased monthly until, after only 6 months from the two release dates, the number of hatchery trout detected was close to zero. The absence of hatchery trout was also confirmed by different electrofishing surveys carried out in the study site. High-water flow events, coupled with other factors, could be the main causes behind the poor effectiveness of stocking actions using subadult/adult fish.

## ACKNOWLEDGEMENTS

We thank the office of hunting and fishing of Canton Ticino, Switzerland, for the data collection, field work and revisions. In particular, we thank Danilo Foresti (Ufficio della caccia e della pesca, Repubblica e Cantone Ticino) and Diego Dagani (BAFU) for the insightful comments.

### Funding

This study was supported by LIFE15 NAT/IT/000823 and Interreg ITA-CH SHARESALMO (Grant No. 599030). The funders had no role in study design, data collection and analysis, decision to publish, or preparation of the manuscript.

### Grant Disclosures

The following grant information was disclosed by the authors:
LIFE15 NAT/IT/000823.
Interreg ITA-CH SHARESALMO: 599030.

### Competing Interests

Armando Piccinini and Richard A. Carmichael are employed by Biomark LLC, Boise, USA.

### Author Contributions

- Stefano Brignone conceived and designed the experiments, analyzed the data, prepared figures and/or tables, authored or reviewed drafts of the article, and approved the final draft.

- Vanessa De Santis analyzed the data, authored or reviewed drafts of the article, and approved the final draft.
- Tiziano Putelli conceived and designed the experiments, performed the experiments, authored or reviewed drafts of the article, and approved the final draft.
- Christophe Molina conceived and designed the experiments, performed the experiments, authored or reviewed drafts of the article, and approved the final draft.
- Armando Piccinini conceived and designed the experiments, performed the experiments, authored or reviewed drafts of the article, and approved the final draft.
- Richard A. Carmichael conceived and designed the experiments, authored or reviewed drafts of the article, and approved the final draft.
- Pietro Volta conceived and designed the experiments, authored or reviewed drafts of the article, and approved the final draft.

## Animal Ethics

The following information was supplied relating to ethical approvals (*i.e.*, approving body and any reference numbers):

Dipartimeto della sanità e della socialità-Esperimenti su animali. Canton Ticino, Switzerland.

## Data Availability

The raw data is available in the Supplemental File.

## Supplemental Information

Supplemental information for this article can be found online at http://dx.doi.org/10.7717/peerj.14069#supplemental-information.

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
