# Peer review of "What’s the effectiveness of stocking actions in small creeks? The role of water discharge behind hatchery trout downstream movement"

_PeerJ, doi:10.7717/peerj.14069_

## Round 0.1 · original submission · Major Revisions

Both reviewers appreciated the work you've put into the paper, but raised a couple major issues. Specifically, (1) they requested more focus around the research question, and (2) do not agree with the results around discharge being supported by data / model output.

Reviewer 1 ·

Basic reporting

In this manuscript, the authors look at the fate of stocked sub-adult/adult trout using fixed antenna system and fish PIT-tagging, and additionally electrofishing. They find that stocked fish quickly disappear from the system, including individuals moving to the lake, especially after high flow events.
As stated to the editor, I have already reviewed a previous version of the MS for Ecohydrology; so this should be taken into account to avoid double jeopardy on the MS.
I find the MS well written, it is professional. I think that the statistical analysis is appropriate and correctly interpreted. I am personally excited to see the results of the study because the data represent some effort and investment to tag and release ~1500 fish. However, I still have two main points:

Experimental design

1. I don’t see so much science novelty. So, I don’t know about the scope of PeerJ and if it fits to the audience. I would personally better see more interest into publishing the study in a more specialist journal with more interested audience.

2. I am still confused about the research question. I like that the authors narrowed the MS to the role of water discharge, but I am not sure what is the take-home message of the study. Basically, most of the stocked fish disappeared from the system (only ~15% of fish redetected at the antenna, and <1% recaptured). So, I actually disagree with the conclusion that “High-water discharge events were probably the main reason behind the steep decrease in hatchery trout abundance over time in our study site” (Abstract). See also last point below.

Validity of the findings

- Method and results: This is great that the authors used wild individuals as comparison. But I don’t see model outputs for the wild population. So, I don’t really see how the comparison is made in the end (but see L.459-467 about method limitation).

- L. 404: “These factors could likely explain why hatchery trout could have been washed downstream after high-water events”. I find this statement a bit partial and I disagree. There are several lines of evidence that suggest that the trout are not “washed away”.
1- It seems that -although not significant-, some fish from downstream actually moved upstream during high flow event.
2- The authors found a 1-day lag between water discharge peak and fish movement, so it is not a physical process.
3- In the previous MS version, we could see the daily distribution of detections. Fish move with sun light cycle, it is an active process.

Additional comments

To conclude, I disagree with interpretation that: water discharge is the main cause of trout disappearance (the data don’t show it), and that it is even responsible for some disappearance. Rather, I think that rain events are the mechanism used by fish to migrate to the lake.

·

Basic reporting

In this paper, the authors examine the effect of high discharge events on the movement rates of hatchery-raised brown trout past an antenna on a small creek. The second stated main goal of this paper was to investigate how water discharge can alter the abundance and residence time of hatchery trout, especially compared to wild trout. However, no data or results were presented in relation to this second goal. The two stated hypotheses are that 1) hatchery trout will be killed or transported downstream during high water events and 2) wild trout are better adapted to high discharge and will have higher survival and lower movement during high water events compared to hatchery fish. While the authors present some support for hypothesis 1 (showing hatchery trout often move downstream after a high water event), there is very little evidence even discussed related to hypothesis 2.

The authors do provide appropriate background discussion and literature references about all topics discussed. The paper is generally well written, however there are a number of sentences or phrases that should be revised to be better comprehended in English. A careful reading by an author or colleague proficient in English could improve the general flow of the paper.

Experimental design

The goals and hypotheses stated in the introduction align well with the aims and scope of PeerJ. The research question and hypotheses are well laid out in the introduction; however I don’t believe they are all sufficiently addressed in the methods, results and discussion.

The paper provides little detail about the electrofishing data or any analysis related to that. The paper fails to mention when electrofishing surveys were carried out, but from examining the data file it appears they occurred at different times of the year compared to hatchery releases. This could confound the comparison between behavior of hatchery and wild trout. No results were presented about the abundance of wild trout based upon this electrofishing, whether there were any recaptures of wild trout, or whether any estimates were made of the efficiency of electrofishing based on recaptures.

I would have liked to see the release of hatchery fish be balanced more between upstream and downstream sites relative to the antenna. The single downstream site appeared to be very close to the mouth of the lake, so perhaps trout moved into the lake and/or up into a different tributary. Also, given the differences in distance between the release sites and the antenna, some differentiation between upstream release sites is called for in the analysis. Either by analyzing fish from each release site separately or incorporating distance from antenna as a covariate somehow. The lag of fish released at site 2, just upstream of the antenna, and fish released at site 4, several km upstream of the antenna will probably be different. As for wild fish, the authors did not present any information as to where the 226 marked wild trout were released.

Validity of the findings

I commend the authors on the use of PIT tags to study fish movement. However, I feel they have not exploited that data to the fullest extent. For example, since they installed two antennas, if the detection efficiency is truly close to 100%, most fish should be detected on both antennas during the same movement event, allowing the researchers to discern the direction of movement. Also, since each detection can be attributed to an individual fish, the researchers could examine whether any fish are detected moving both upstream and downstream, or how often an individual fish crosses the antenna site.

The authors’ general conclusion, that high discharge events lead to the disappearance of hatchery trout from this creek, is not necessarily supported by the evidence they present. A single antenna site can only measure movement across that location. The lack of detections after 6 months does not necessarily mean hatchery fish have “disappeared”, it could mean they have stopped moving. The electrofishing surveys could help determine which of those outcomes is more likely, but it is unclear whether those surveys actually captured most of the fish in the system (if that was true, then given the low recapture rate that would suggest that most hatchery fish had left), or were just very inefficient.

The authors’ claimed to compare movement and survival rates between hatchery and wild trout, but no such comparisons were actually made with the data they collected. They provided several references describing previously measured differences between hatchery and wild fish, but I believe they need to flesh out more from their data to speak to these comparisons.

Finally, the authors do not address the idea that hatchery fish may prefer the lake habitat, and preferentially move downstream towards that habitat. Rain events may aid them in this migration, not “force” them downstream.

Additional comments

The abstract is missing a few prepositions or the wrong one was used. L 19 should be “of the effectiveness”

L244: should be “variation in fish numbers detected over time”. But I don’t think describing the cross-correlation analysis as testing whether variation of NFDD is aligned with variation of water discharge is correct, that sounds like periods of high variance of NFDD would coincide with periods of high variance of water discharge.

L323-324: I would rephrase the first sentence, as I don’t believe that most hatchery fish “disappeared”. I think the authors are referring to fish not being detected, which is more likely due to the fish settling into a particular reach and not migrating past the antenna any longer. Similar comment to L358, “fast disappearance”, and few other places in the discussion.

L327-330: Along the same lines, the authors’ conclusion that stocking with subadult/adult hatchery trout (which is illegal anyways) should be considered with caution is not supported by the data. The fact that 6 months after release no hatchery fish were detected on the antenna does not mean they have all died or left the system; it could be that they are simply not migrating.

L358-362: The electrofishing methodology also has less than perfect detection efficiency, so the fact that only 13 marked (hatchery?) trout were recaptured could be due to poor efficiency.

I appreciate the author sharing their data. However, it would be appropriate to include a tab in the Excel workbook providing some metadata to allow others to better understand what exactly that data are. In addition, the format of dates is not consistent within the data (sometimes month/day/year, sometimes day/month/year).

---

## Round 0.2 · accepted · Accept

Thanks for addressing the reviewers comments in this revision -- the paper is substantially improved, nice work!